# Protein Kinase C Regulates Expression and Function of the Cav3.2 T-Type Ca^2+^ Channel during Maturation of Neonatal Rat Cardiomyocyte

**DOI:** 10.3390/membranes12070686

**Published:** 2022-07-02

**Authors:** Yan Wang, Masaki Morishima, Katsushige Ono

**Affiliations:** 1Department of Pathophysiology, Oita University School of Medicine, Oita 879-5593, Japan; wang@oita-u.ac.jp (Y.W.); mmoris@nara.kindai.ac.jp (M.M.); 2Department of Cardiology and Clinical Examination, Faculty of Medicine, Oita University, Oita 870-1192, Japan; 3Department of Food Science and Nutrition, Kindai University Faculty of Agriculture, Nara 631-8505, Japan

**Keywords:** T-type Ca^2+^ channel, Cav3.1, Cav3.2, PKC, Nkx2.5, chelerythrine

## Abstract

Two distinct isoforms of the T-type Ca^2+^ channel, Cav3.1 and Cav3.2, play a pivotal role in the generation of pacemaker potentials in nodal cells in the heart, although the isoform switches from Cav3.2 to Cav3.1 during the early neonatal period with an unknown mechanism. The present study was designed to investigate the molecular system of the parts that are responsible for the changes of T-type Ca^2+^ channel isoforms in neonatal cardiomyocytes using the whole-cell patch-clamp technique and mRNA quantification. The present study demonstrates that PKC activation accelerates the Ni^2+^-sensitive beating rate and upregulates the Ni^2+^-sensitive T-type Ca^2+^ channel current in neonatal cardiomyocytes as a long-term effect, whereas PKC inhibition delays the Ni^2+^-sensitive beating rate and downregulates the Ni^2+^-sensitive T-type Ca^2+^ channel current. Because the Ni^2+^-sensitive T-type Ca^2+^ channel current is largely composed of the Cav3.2-T-type Ca^2+^ channel, it is accordingly assumed that PKC activity plays a crucial role in the maintenance of the Cav3.2 channel. The expression of Cav3.2 mRNA was highly positively correlated with PKC activity. The expression of a transcription factor Nkx2.5 mRNA, possibly corresponding to the Cav3.2 channel gene, was decreased by an inhibition of PKCβII. These results suggest that PKC activation, presumably by PKCβII, is responsible for the upregulation of Ca_V_3.2 T-type Ca^2+^ channel expression that interacts with a cardiac-specific transcription factor, Nkx2.5, in neonatal cardiomyocytes.

## 1. Introduction

Voltage-gated calcium (Ca^2+^) channels govern the rapid entry of Ca^2+^ ions into a wide variety of excitable cells and are involved in both electrical and cellular signaling. In cardiac myocytes, two distinct families of voltage-gated Ca^2+^ channels, high-voltage-activated L-type Ca^2+^ channels and low-voltage-activated T-type Ca^2+^ channels, are identified. The L-type Ca^2+^ channel current (I_Ca.L_) is well characterized and is known to play a central role in the excitation-contraction coupling mechanism. In contrast, the T-type Ca^2+^ channel current (I_Ca.T_) is known to contribute to cellular automaticity [1]. Because I_Ca.T_ is activated at more negative voltages, resulting in Ca^2+^ entry-dependent membrane slow depolarization, it is also postulated that the excessive Ca^2+^ overload and the resulting arrhythmias may be related to the abnormal function of the T-type Ca^2+^ channels [2]. Three genes encode the pore-forming α_1_ subunits of T-type Ca^2+^ channels, Ca_V_3.1, Ca_V_3.2 and Ca_V_3.3. Among them, cardiomyocytes express two isoforms of the T-type Ca^2+^ channels, Ca_V_3.1 and Ca_V_3.2. These isoforms differ in their sensitivity to nickel (Ni^2+^); the Ca_V_3.2 channels are blocked at 20-fold lower concentrations than Ca_V_3.1 [3].

In the heart, T-type Ca^2+^ channels or I_Ca,T_ are robustly observed embryonic or neonatal cultured atrial and ventricular myocytes, whereas they are drastically reduced in normal adult ventricular myocytes [2]. Changes in their expression levels during the development suggest an important role for the T-type Ca^2+^ channels at specific stages of the fetal and neonatal cardiomyocytes. A quantitative analysis of the pore-forming α_1_ subunit of the T-type Ca^2+^ channels revealed that the levels of Ca_V_3.2 mRNA are high in embryonic cardiac tissue and at 3 weeks postnatal but become undetectable at 5 weeks [4]. In contrast, the levels of Ca_V_3.1 mRNA are maintained in cardiomyocytes from the fetal stage to adulthood; they are restricted in nodal cells and Purkinje cells in adults. Although changes in the T-type Ca^2+^ channels’ expression and isoform switch could be important for the development of the differentiated myocytes phenotypes, the molecular mechanisms that regulate the expression of individual T-type Ca^2+^ channel genes and the functional consequences of alterations in the T-type Ca^2+^ channel activity during development remain to be clarified. 

Protein kinase C (PKC) is an enzyme in signal transduction that is involved in a variety of cellular functions and has been identified as a key molecule in various pathological conditions in the heart [5]. Three PKC subgroups have been identified: (1) conventional PKC (α, βI, βII, and γ), which are activated by phosphatidylserine, intracellular Ca^2+^ and diacylglycerol (DAG); (2) novel PKC (δ, ε, η and θ), which are activated by DAG but not intracellular Ca^2+^; (3) atypical PKC (ζ and λ/ι), which are not activated by intracellular Ca^2+^ or DAG [5]. Out of the 11 PKC isoforms, α, βI, βII, δ, ε, ζ and λ have been identified in the heart [5,6,7,8]. PKC may phosphorylate a myriad of target proteins including transcription factors and their modulators, thereby regulating transcriptional processes [9]. 

Although our knowledge of stem cell biology has recently advanced to the point where we now can induce cardiomyocyte-like cells, it still remains incompletely understood how molecular signals can regulate ion channel differentiation or maturation. For the purpose of regenerative medicine for instance, many researchers have been committed to improving the maturation of cardiomyocytes, such as in induced pluripotent cells (iPS). Thus, understanding the mechanisms regulating ion channel maturation will provide significant insights into the directed electrical maturation of the heart and in improving the knowledge on cardiomyocyte differentiation. The present study is therefore designed to investigate the mechanisms that regulate the expression of Ca_V_3.1 and Ca_V_3.2 genes during the differentiation of myocytes in the rat neonatal heart. Specifically, the purpose of the current study is to identify the molecular mechanism that governs the isoform switching of the T-type Ca^2+^ channels from the Cav3.2 channel to the Cav3.1 channel in the cardiomyocytes during the perinatal phase, using the whole-cell patch-clamp technique and mRNA quantification. Our results indicate that PKC activation, presumably by PKCβII, upregulates Ca_V_3.2 T-type Ca^2+^ channel expression that interacts with a cardiac-specific transcription factor, Nkx2.5, which also suggests a novel mechanism for the T-type Ca^2+^ channel remodeling in pathological conditions of the heart.

## 2. Materials and Methods

### 2.1. Preparation of Neonatal Rat Cardiomyocytes and Cell Culture

Neonatal rat cardiomyocytes were prepared as described previously [10,11]. Neonatal ventricular myocytes were prepared from 2-8-days-old Wistar rats and maintained at 37 °C under 5% CO_2_ in Dulbecco’s modified Eagle’s medium (DMEM) that was supplemented with 10% fetal bovine serum for 24 h, followed by an additional 24 h incubation in DMEM with or without protein kinase inhibitor, or a protein kinase activator (see the figure insets). The cardiomyocytes were washed with vehicle for 30 min to avoid acute actions of PKC activator or inhibitors on I_Ca.T_ prior to the electrophysiological examination. 

### 2.2. Stable Cell Lines for Cav3.1 and Cav3.2 Channels

Cav3.1 and Cav3.2 isoforms derived from human heart, which forms cardiac T-type Ca^2+^ channels, was stably expressed in human embryonic kidney (HEK)-293 cells with no auxiliary subunits; Cav3.1- and Cav3.2-HEK cells. These cell lines were generously gifted by Prof. Perez-Reyes. A profile and a procedure for channel expression were described in detail in a previous report [12]. The Cav3.1 and Cav3.2-HEK cells were maintained in DMEM, supplemented with 10% fetal calf serum, 100 U/mL of penicillin, and 100 μg/mL of streptomycin in an atmosphere of 95% O_2_ plus 5% CO_2_ at 37 °C. This medium was supplemented with 300 g/mL of G418 (neomycin analogue) for the selection of recombinant HEK-293 cells [12].

### 2.3. Electrophysiology

For the electrophysiological recordings, we used a standard whole-cell patch clamp technique throughout the study [5,13]. The voltage clamp mode was used to measure the ionic currents, and the current clamp mode was used to measure the action potentials using EPC-9 (HEKA Elektronik, Lambrecht, Germany). The temperature of the external solution was kept at 37 °C using a chamber heating system (Bipolar Temperature Controller, model TC-202A, Harvard Apparatus, Holliston, MA, USA). Patch pipettes (2 to 3 MΩ electrical resistance, filled with the pipette solutions described below) were pulled from micro glass capillaries (Drummond, Broomall, PA, USA) using the Micropipette Puller, Model P-97 (Sutter Instrument, Novato, CA, USA). Series resistance was electronically compensated as much as possible without oscillation (60 to 75%). Capacitive artifacts were minimized by using the built-in circuitry of the amplifier. The remaining transients and linear leakage currents were eliminated by using p/4 subtraction (Pulse/Pulsefit, HEKA Elektronik). The amplifier output was cut-off filtered at 5 kHz, digitally sampled at 10 kHz by using an ITC-16 interface (InstruTech Corp., Great Neck, NY, USA), and stored on a computer under the control of a data acquisition program (Pulse/Pulsefit, HEKA Elektronik). For continuous action potential recording, the amplifier output was sampled using Power Lab (AD Instruments, Sydney, Australia) and stored on a computer using Chart software (AD Instruments). I_Ca.L_ was initially recorded from the holding potential (V_HP_) of −50 mV, followed by various test potentials. I_Ca.T_ was then recorded by subtracting the currents that were obtained by V_HP_ of −50 mV from those that were obtained by V_HP_ of −120 mV in the same patch. All the experiments were conducted at 37 °C.

### 2.4. Solutions and Chemicals

For the action potential recordings (current clamp recording), the bath solution (Tyrode’s solution) contained (mM): NaCl 140; MgCl_2_ 1; KCl 5.4; HEPES 10; glucose 10; and CaCl_2_ 1.8 (pH of 7.4 adjusted with 1 mol/L of NaOH), and the pipette solution contained (mM): KCl 140; MgCl_2_ 2; creatine phosphate 5; HEPES 10; EGTA 0.05; and Mg-ATP 5 (pH of 7.2 adjusted with 1 mol/L of KOH). For measuring I_Ca.T_, Tyrode’s solution was supplemented with 0.03 mM of tetrodotoxin to eliminate the Na^+^ current, and the pipette solution contained (mM): CsCl 130; MgCl_2_ 2; Mg-ATP 2; Na_2_-GTP 2; EGTA 10; and HEPES 5 (pH of 7.2 adjusted with 1 mol/L of CsOH). The PKC activator or inhibitors were not included in the bath solution. A PKC activator PMA (phorbol 12-myristate 13-acetate); a pan-protein kinase C (PKC) inhibitor chelerythrine (1,2-dimethoxy-N-methyl(1,3)benzodioxolo(5,6-c) phenanthridinium chloride); a PKCα inhibitor Ro-32-0432 (3-(8-((dimethylamino)methyl)-6,7,8,9-tetrahydropyrido [1,2-*a*]indol-10-y1)-4-(1-methyl-1*H*-indol-3-y1)-1*H*-pyrrole-2,5-dione hydrochloride); a PKCα inhibitor Gö 6976 (12-(2-Cyanoethyl)-6,7,12,13-tetrahydro-13-methyl-5-oxo-5H-indolo(2,3-a) pyrrolo(3,4-c)-carbazole); and a PKCβI inhibitor CGP41251 4′-N-benzoyl staurosporine, and a PKCβI/PKCβII inhibitor 3-anilino-4-[1-(3-imidazol-1-ylpropyl)indol-3-yl]pyrrole-2,5-dione, IYIAP) were purchased from Calbiochem Co. (La Jolla, CA, USA). These protein kinase inhibitors were applied at the concentration of 5-10 times the ones inducing a 50% inhibition (IC_50_) of each protein kinase [14,15,16,17]. These protein kinase inhibitors were dissolved in dimethyl sulfoxide (DMSO), where the final concentration of DMSO was less than 0.01%. 

### 2.5. Quantitative Real-Time PCR

The total RNA was extracted from rat neonatal myocytes using Isogen (Nippon Gene, Tokyo, Japan) 48–60 h after adenoviral infection. The cDNA was synthesized from 1 μg of total RNA using a Transcriptor First Strand cDNA Synthesis kit (Roche Molecular System Inc., Alameda, CA, USA). The real-time PCR was performed on Light Cycler (Roche) using the FastStart DNA Master SYBR Green I (Roche) as a detection reagent. Forward and reverse primer sequences, respectively, for the rat Ca^2+^ channel subunits and transcription factors were designed from their sequence in the GenBank database as follows (accession numbers are indicated in parentheses): Ca_v_3.1 (AF027984), forward 5′-TCT CTA GGG CTA TAG GCG-3′, reverse 5′-GGA GAT TTT GCA GGA GCT AT-3′; Ca_v_3.2 (AF290213), forward 5′-GGC GAA GAA GGC AAA GAT GA-3′, reverse 5′-GCG TGA CAC TGG GCA TGT T-3′; Nkx2.5 (NM_053651), forward 5′-ATC GCG GTG CCG GTG TT-3′, reverse 5′-GCC CGA ATT GCC CTG TG-3′; GAPDH (M17701), forward 5′-GCC ATC AAC GAC CCC TTC AT-3′, reverse 5′-TTC ACA CCC ATC ACA AAC AT-3′. Data were calculated by 2^−ΔΔCT^ and were presented as the fold change that was induced in the transcript of each myocyte gene that was assayed by ethanol exposure. The gene expression was normalized to that of GAPDH and compared with the control condition (defined as 100%). The size of the PCR products was confirmed using 2% agarose gel electrophoresis.

### 2.6. Preparation of Recombinant Adenoviruses 

The production and confirmation of the ability of recombinant adenovirus expressing a transcription factor Nkx2.5 (Ad-Nkx2.5) was made according to the previous study [18]. In short, the recombinant adenoviruses encoding Nkx2.5 were generated using the E1-deleted pAdEasy-1 adenoviral plasmid and the shuttle vector, pAdTrack-CMV, containing green fluorescent protein (GFP). For the control group, the virus vector expressing only GFP (Ad-GFP) was constructed by the same technique [18].

### 2.7. Measurement of PKC Activation 

Membranous PKC activity was measured using the StressXpress PKC kinase activity assay kit (EKS-420A; Stressgen Bioreagents Corp., Victoria, BC, Canada). This assay is a solid-phase ELISA that uses a specific synthetic peptide as a substrate for PKC and a polyclonal antibody that recognizes the phosphorylated form of the substrate.

### 2.8. Action Potential Parameters

The action potential parameters were measured based on the off-line digital data with or without digital differential meters using computer software (SigmaPlot14, Sigma-Aldrich, St. Louis, MO, USA). The take-off potentials (TOP) were measured in action potential recordings at the time of the maximum values of the second derivative of the membrane potential with respect to time (d^2^V/dt^2^). The diastolic depolarization rate was determined as the slope between the maximum diastolic potential point and the take-off potential point. 

### 2.9. Data Acquisition and Statistical Analysis

The data were acquired by using computer software (Pulse/PulseFit, V.8.11, HEKA Elektronik). The group data are shown as means ± S.D. if nothing else is stated. An analysis of variance and Tukey–Kramer procedure were used for multiple comparisons, and the Student’s *t* test was used for the comparison of the two groups. Differences were considered significant when *p* values were less than 0.05.

The relationship between Ni^2+^ concentration and its inhibitory effect on I_Ca.T_ was fitted in a nonlinear fashion using SigmaStat 3.5 software (Sigma-Aldrich). The equation was Y = Bottom + (Top − Bottom)/(1 + 10((LogIC_50_ − X)*Hill Slope)), where Hill Slope represents the steepness of the family of curves, Top and Bottom represent plateaus in the units of the y-axis, X represents the logarithm of concentrations of Ni^2+^, and Y represents the normalization current. 

## 3. Results

### 3.1. Nickel as a Selective Inhibitor of Cav3.2 T-Type Ca^2+^ Channel

Although Ni^2+^ is widely accepted to block the T-type Ca^2+^ channel with subtype selectively, we first reevaluated the efficacy of Ni^2+^ to inhibit the Cav3.1 and Cav3.2 T-type Ca^2+^ channel in the heterologous expression system, because our electrophysiological study largely depended on the distinct blocking ability of Ni^2+^ on the Cav3.2 channel in comparison with those on the Cav3.1 channel. As partially expected by the literature, Ni^2+^ preferably inhibited the Cav3.2 channel with an IC_50_ value of 5.7 μM, rather than the Cav3.1 channel (IC_50_ value of 171 μM) (Figure 1), which indicates that Ni^2+^ is a potential discriminator that is suitable to separate the Cav3.2 channel current (Cav3.2-I_Ca.T_) from whole I_Ca.T_ in cardiomyocytes. In this context, Ni^2+^-sensitive I_Ca.T_ could be assumed to represent Cav3.2-I_Ca.T_. 

### 3.2. Cellular Automaticity in Neonatal Cardiomyocytes

Neonatal ventricular cardiomyocytes are distinctly characterized by the cellular automaticity, which is obviously distinguished from normal adult ventricular cardiomyocytes. Cellular automaticity or spontaneous beating ability is detectable and recordable in neonatal rat cardiomyocyte, although the beating rate is much slower than the sinus node-driven heart rate (Figure 2). The spontaneous beating rate was significantly decreased when the cardiomyocytes were treated by a PKC inhibitor, chelerythrine, for 24 h (Figure 2B,E). On the other hand, the spontaneous beating rate was highly maintained when the cardiomyocytes were exposed to a PKC activator, phorbol 12-myristate 13-acetate (PMA), for 24 h (Figure 2C,F). Although the slow diastolic depolarization rate (DDR) tended to decrease after the birth (Figure 2H), chelerythrine and PMA failed to significantly modify DDR. The maximum dV/dt of the action potential was significantly increased, and the take-potential (TOF) was more depolarized when the cardiomyocytes were matured at day 8 in comparison to those in day 2 (Figure 2I and Figure 3K). Other action parameters including the maximum diastolic depolarization (MDP) and overshoot of the action potentials were unchanged during the observation period (Figure 2G,J). As expected by normal maturation steps in ventricular cardiomyocytes, cellular automaticity gradually declines after the birth, as assessed by the current clamp recordings of membrane potentials (Figure 3). We noted that the Ni^2+^-resistant component of the beating rate was substantially unchanged for up to 8 days after the birth, and that it was not affected by PMA or chelerythrine (Figure 3A–C). On the other hand, the Ni^2+^-sensitive component of the cardiomyocytes’ beating rate remained high at day 8 after the birth when PKC activity was activated by PMA that was applied to the cell culture medium (Figure 3C). The facts that PKC activation upregulated the beating rate, whereas PKC inhibition downregulated the beating rate through the examination protocol from day 2 to day 8 (Figure 3D) indicated a role of PKC activity for the cell beating. We also noted that an age-dependent reduction in the Ni^2+^-sensitive beating rate was highly maintained when a PKC activator PMA was applied for 24 h (Figure 3E). Since Ni^2+^ highly favorably inhibits the Cav3.2 T-type Ca^2+^ channel (Figure 1), and because the T-type Ca^2+^ channels have been suggested to play a critical role in cardiac automaticity, it is accordingly suggested that T-type Ca^2+^ channel-dependent reduction in spontaneous beating, particularly the Cav3.2 T-type Ca^2+^ channel, is possibly involved in the changes of PKC activity in neonatal cardiomyocytes.

### 3.3. Ni^2+^-Sensitive and -Insensitive T-Type Ca^2+^ Current in Neonatal Cardiomyocytes

I_Ca.T_ density is appreciably high in neonatal cardiomyocytes, as has been previously reported [19]; the maximum inward I_Ca.T_ was 7.07 ± 0.64 pA/pF (*n* = 8) at day 2 and 3.24 ± 0.34 pA/pF (*n* = 8) at day 8 (Figure 4C). I_Ca.T_ density was gradually decreased after the birth, which was very similar to that of the spontaneous beating rate in terms of the time (age) dependency (Figure 3). More importantly, it is noteworthy that I_Ca.T_ was highly maintained when the cardiomyocytes were exposed to PMA, and conversely I_Ca.T_ was significantly decreased when the cardiomyocytes were treated by a PKC inhibitor, chelerythrine (Figure 4C). Importantly, PMA and chelerythrine were washed out prior to the electrophysiological study. Thus, the acute actions of PKC for the channel phosphorylation could be minimized in this experimental protocol. Accordingly, the actions of PMA or chelerythrine may reflect their long-term effects for 24 h on the electrophysiological properties in cardiomyocytes. Interestingly, Ni^2+^-insensitive or Ni^2+^-resistant I_Ca.T_ was unchanged throughout the protocol period up to 8 days, regardless of the presence of PMA or chelerythrine (Figure 4C), which suggests that an age-dependent decline of I_Ca.T_ could be attributed to an age-dependent decline of Ni^2+^-sensitive I_Ca.T_. In fact, at day 2, Ni^2+^-sensitive I_Ca.T_ (vehicle) was 4.90 ± 0.42 pA/pF, whereas it was significantly reduced to 1.25 ± 0.23 pA/pF at day 8 (Figure 4D). Although Ni^2+^-sensitive I_Ca.T_ in the PMA-treated cardiomyocyte was declined during the observation period for 8 days, it was highly maintained at day 8 (3.96 ± 0.26 pA/pF), which was equal to 317% of I_Ca.T_ at day 8 in vehicle (Figure 4C). Conversely, Ni^2+^-sensitive I_Ca.T_ in a chelerythrine-treated cell was much smaller than that in vehicle; I_Ca.T_ at day 2 in a chelerythrine-treated cell was 2.86 ± 0.32 pA/pF, which was equal to 58% of I_Ca.T_ treated in vehicle; and I_Ca.T_ at day 8 in a chelerythrine-treated cell was 0.73 ± 0.17 pA/pF, which was equal to 58% of I_Ca.T_ treated in vehicle on the same day (Figure 4D). We would like to emphasize that a low concentration of Ni^2+^ preferably inhibits the Ca_V_3.2 T-type Ca^2+^ channel but not the Ca_V_3.1 T-type Ca^2+^ channel (Figure 1). Therefore, the results with a time-dependent decline of Ni^2+^-sensitive I_Ca.T_ and its sensitivity to PKC led us to consider that PKC activity has a role in maintaining Ca_V_3.2-I_Ca.T_ after the birth in neonatal cardiomyocytes. Meanwhile, the cell capacitance tended to be larger in the cardiomyocytes at day 8 (22.9 ± 2.6 pF in vehicle) in comparison with those at day 2 (16.7 ± 1.8 pF in vehicle), although there was no significant difference among the groups, regardless of the presence or absence of a PKC inhibitor/activator.

### 3.4. PKC Upregulates Ca_V_3.2 mRNA

To the best of our knowledge, among two types of the T-type Ca^2+^ channel in the heart, Ca_V_3.1 and Ca_V_3.2, the latter could be more susceptible to being affected by cellular maturation. In order to confirm this, we tested for the possible changes of mRNAs coding Ca_V_3.2 genes that were derived from cardiomyocytes at day 2, 4, 6 and 8 by performing an RT-PCR (Figure 5). Consistently with the electrophysiological studies in Figure 4D, mRNA for Ca_V_3.2, an Ni^2+^-sensitive T-type Ca^2+^ channel isoform, was significantly smaller in the chelerythrine-treated cells, whereas it was significantly larger in PMA-treated cells in comparison with those in vehicle-treated cells (Figure 5A). In the same way, mRNA for Ca_V_3.2 was significantly smaller in the chelerythrine-treated cells, whereas it was significantly larger in the PMA-treated cells than those in the vehicle-treated cells at day 4 (Figure 5B). A plot of Cav3.2-mRNA expression ratio against time (day after birth) reveals a time-dependent decline of Cav3.2 mRNA (Figure 5C), which is very similar to changes in Ni^2+^-sensitive I_Ca.T_ that are shown in Figure 4D. Importantly, chelerythrine downregulated the amount of Cav3.2-mRNA, and PMA upregulated the amount of Cav3.2-mRNA. These Cav3.2 mRNA modulations were significantly evident through the observation period up to 8 days after the birth (Figure 5C). Intriguingly, chelerythrine not only significantly reduced Cav3.2-mRNA in comparison with PMA or vehicle, but also accelerated the age-dependent decline of it after the birth (Figure 5C). Considering that Ni^2+^-insensitive I_Ca.T_, largely composed of the Cav3.1 T-type Ca^2+^ channel remains unchanged after the birth (Figure 4C), and that PKC activity-sensitive Cav3.2-mRNAappreciably declines after the birth, it becomes clear that a time-dependent switching of the dominant T-type Ca^2+^ channel isoform from Ca_V_3.2 to Ca_V_3.1 is firmly associated with the PKC activity during the perinatal period of rat development. Then, the important question is, how likely is it that Cav3.2 expression is associated with PKC activity? To answer this question, we determined the strength of a correlation between the expression levels of Cav3.2 and the PKC activity (Figure 6).

The distribution of the two variables, PKC activity and the expression levels of Cav3.2 mRNA, are obvious positively correlated with each other in vehicle (Figure 6A), under the effect of chelerythrine (Figure 6B), and under the effect of PMA (Figure 6C), although two variables were highly distributed under the effect of PMA, and they were lowly distributed under the effect of chelerythrine. There was a significant linear relation between PKC activity and the expression levels of Cav3.2 mRNA in the condition that data were all included (Figure 6D); the coefficient of determination, Rsqr, was calculated as equal to 0.828, suggesting a very close relation between PKC-dependent signals for Cav3.2 gene expression. 

### 3.5. Cav3.1 Expression Is Negatively Associated with PKC Activity

In comparison with a plot of Cav3.2 mRNA expression ratio against time, we measured Cav3.1 mRNA expression in the cardiomyocytes of rats from day 2 to day 8 after the birth, cultured with or without PMA (Figure 7). The expression of Cav3.1 mRNA was much lower than that of Cav3.2 mRNA, which was unchanged with age (day after birth). Contrary to the effect on Cav3.2 mRNA, PMA downregulated the expression of Cav3.1 mRNA. As a result of this change, the Cav3.2-mRNA/Cav3.1-mRNA ratio was highly maintained by PKC activation with PMA. Although PMA was not able to halt the age-dependent decline of Cav3.2 expression completely, PMA successfully maintained the Cav3.2-mRNA/Cav3.1-mRNA ratio above 10 for up to 8 days after the birth. 

### 3.6. Transcription Factor Nkx2.5 Is Responsible for PKC Actions for Cav3.2

Cardiac-specific transcription factor Nkx2.5 has previously been identified to possibly regulate the expression of the Ca_V_3.2- but not the Cav3.1-T-type Ca^2+^ channel [18]. In this study, we have constructed recombinant adenovirus expressing a transcription factor Nkx2.5 (Ad-Nkx), in order to investigate the molecular mechanisms during the isoform switching. By the overexpression of Nkx2.5 in neonatal cardiomyocytes, I_Ca.T_ was highly increased as compared with the myocytes that were treated in vehicle and the GFP-overexpressed myocytes (Figure 8A). An analysis of the group data revealed a significant increase in I_Ca.T_ in the neonatal cardiomyocytes that were treated by Ad-Nkx for 24 h (Figure 8B). We then performed quantitative analyses of mRNA revels for Nkx2.5 by use of a real-time PCR to investigate the actions of PKC on the expression of Nkx2.5. In Figure 8C, the level of Nkx2.5 mRNA was significantly decreased by a pan-PKC inhibitor, chelerythrine. To further investigate the subtype specific actions of PKC, we employed a specific inhibitor of PKCα (Gö 6976 and Ro-32-0432); specific inhibitors of PKCβI (CGP41251 and IYIAP of 5 nM); and a PKCβII inhibitor (IYIAP of 50 nM) in this context. Interestingly, the expression of Nkx2.5 mRNA was unaffected by PKCα inhibitors (Gö 6976 and Ro-32-0432) or by PKCβI inhibitors (CGP41251 and IYIAP of 5 nM). On the contrary, a high concentration (50 nM) of IYIAP that inhibits both PKCβI and PKCβII significantly reduced the expression of Nkx2.5 mRNA, suggesting a PKCβII-dependent signal pathway for Nkx2.5 expression possibly involving the transcription of the Ca_V_3.2 gene.

## 4. Discussion

The present study demonstrates that PKC activation accelerates the Ni^2+^-sensitive beating rate and upregulates Ni^2+^-sensitive I_Ca.T_ in neonatal cardiomyocytes as a long-term effect. On the other hand, PKC inhibition delayed the Ni^2+^-sensitive component of beating rate and downregulated Ni^2+^-sensitive I_Ca.T_. Because Ni^2+^-sensitive I_Ca.T_ is largely composed of Cav3.2-I_Ca.T_, it is accordingly assumed that PKC activity plays a crucial role in the maintenance of the Cav3.2 channel. Although the Cav3.2 T-type Ca^2+^ channel declined in neonatal cardiomyocytes immediately after the birth, it was significantly highly maintained by PKC activation. Based on the findings that transfection of a transcription factor, Nkx2.5, into cardiomyocytes upregulated I_Ca.T_ and cellular automaticity [18], and that PKC inhibitors downregulated the expression of Nkx2.5 mRNA, it is suggested that the predominant isoform of the T-type Ca^2+^ channel Ca_V_3.2 in cardiomyocytes at the neonatal stage is fate-to-fade, corresponding with a reduction in or cessation of intensive PKC activity that is possibly caused by endogenously or maternally employed serum-associated signals during pregnant periods. 

### 4.1. Isoform Switching of the T-Type Ca^2+^ Channels in the Neonatal Heart

It is widely recognized that the Cav1.2-L-type Ca^2+^ channel and the Cav3.2-T-type Ca^2+^ channel, but not the Cav3.1-T-type Ca^2+^ channel, play crucial roles in the genesis of myocardial electrical activity at the early embryonic stage when cardiomyocytes start self-berating [20,21]. It is also generally accepted that the Cav3.1 channel is mostly responsible for the function of the T-type Ca^2+^ channel in the mid- to late-gestational fetal myocardium [22]. Accordingly, switching of the T-type Ca^2+^ channel isoforms from Cav3.2 to Cav3.1 during the developing stage of embryonic/fetal myocardium is highly plausible. Several studies have described developmental changes of the T-type Ca^2+^ channel isoforms in the embryonic heart or cardiac precursor cells derived from embryonic stem cells [21,23,24], although the mechanism involved is not known. To identify the molecular background behind the phenotypic electrical maturation, this investigation was designed to unveil the long-term effect of PKC on the expression of the T-type Ca^2+^ channel isoforms. This study successfully demonstrates a significant positive correlation between PKC activity and the expression of Cav3.2 mRNA. Because PKC activity gradually declined in the heart after the birth, we postulate that the reduction in the Cav3.2 channel following the developmental maturation of cardiomyocytes was a consequence of the lessened PKC activity in cardiomyocytes. Of note, Mizuta et al. reported that Nkx2.5-positive cells abundantly expressed Cav3.2 mRNA during cardiac differentiation [21]. This notion is consistent with the present result that PKC activity-dependent expression of Nkx2.5 mRNA upregulated I_Ca.T_ (Figure 8). Then, what is the key molecule that is responsible for the active PKC pathway in fetal/neonatal cardiomyocytes? Since it is increasingly evident that progesterone activates kinase pathways, including PKCs, to have multiple downstream actions in a wide variety of cells [25], highly maintained progesterone in maternal blood may account for the mechanism that is involved in the upregulation of the Cav3.2 channel in the fetal heart and its decline after the birth. Interestingly, estrogen is known to downregulate the Cav3.2 channel by the suppression of Nkx2.5 in neonatal cardiomyocytes [26] as a long-term effect. Other maternal serum factors including gonadotropin or gonadotropin-related oligopeptides may also affect PKC activity in cardiomyocytes at the early neonatal stage. Further investigation is needed to clarify the maternal factors that are responsible for PKC activity and its impact on the transcription factors that govern myocardial maturation in the fetal heart.

### 4.2. Actions of PKC for the Cac3.2 Channel

PKC plays a central role in the regulation of cell differentiation, growth and transformation in the heart [27]. Members of the PKC family are divided into three subfamilies on the basis of their mechanisms of activation: the classical PKCs (α, β1, β2, γ); novel PKCs (δ, ε, η, θ); and atypical PKCs (ζ, λ/ι) [28]. Although nearly all PKC isoforms reportedly act in the cardiovascular system, it is particularly important to note that PKC-α, PKC-β, PKC-ε, and PKC-ζ expressions are high in fatal and neonatal hearts but decrease in expressions in adult hearts [29]. In this context, it could be postulated that the abundant expression of the Cav3.2-T-type Ca^2+^ channel in neonatal cardiomyocytes and its age-dependent decline after the birth could be attributed to the developmental decline of PKCβ actions in the neonatal period, because the expression of Cav3.2 mRNA is highly correlated with PKC activity (Figure 6D), and because a PKCβII inhibitor, IYIAP, significantly decreases the expression of NKx2.5 mRNA (Figure 8C). PMA is the most widely used diacylglycerol mimetic agent and inducer of PKC-mediated cellular responses in biological investigations [30]. Since PMA, applied for 24 h, accelerated the beating, increased I_Ca.T_ density and increased Cav3.2 mRNA in cardiomyocyte, the diacylglycerol-dependent regulation of the Cav3.2-T-type Ca^2+^ channel in neonatal cardiomyocyte was firmly confirmed. The conclusion was also verified by a very close positive relation between PKC activity and Cav3.2 mRNA expression. Interestingly, as we have previously reported, PKC activation increases Cav3.2-I_Ca.T_ as a short-term effect in cardiomyocyte, possibly through PKCα actions [5]. In this study, upregulation of the Cav3.2 channel by PMA was confirmed as possibly occurring through PKCβ actions. Taken together, it is of particular interest that there are synergistic short-term and long-term positive impacts of PKC actions on the Cav3.2 channel. In fact, the activation of PKC triggers a multitude of pathophysiological processes in the diseased heart, including heart failure and hypertrophy [27]. Although the T-type Ca^2+^ channels are not expressed in adult ventricular myocytes in normal condition, unusual expression of the T-type Ca^2+^ channel has been found in hypertrophied and failing ventricular cardiomyocytes [2,27]. In this context, this investigation shed light on the relation between PKC activities and the abnormal expression/function of the T-type Ca^2+^ channels in the heart. Although we validated that the inhibition of PKCβII activity reduced the expression of Nkx2.5, which is known to regulate the transcription of the Cav3.2 gene, we could not further ascertain the interaction molecules between PKCβII and Nkx2.5. Further investigations to clarify the signal cascades are obviously needed. 

### 4.3. Study Limitation

Multiple limitations of the current study should be acknowledged: (1) This was an experimental study by use of rodent, and the results may not be extrapolated to human subjects, such as in human iPS studies; (2) the regulatory signal pathways for Ni^2+^-insensitive I_Ca.T_, mostly composed of Cav3.1, were unidentified; (3) a protein-level analysis for Cav3.1 and Cav3.2 was not performed; (4) possible changes of PKC isoform-specific activity were not assessed. These questions should be addressed thoroughly in future studies.

## Figures and Tables

**Figure 1 membranes-12-00686-f001:**
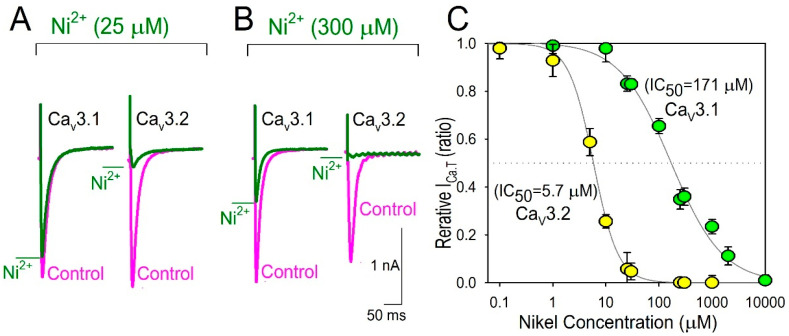
Effect of Ni^2+^ on the Cav3.1 channel and the Cav3.2 channel. Representative Cav3.1- and Cav3.2-T-type Ca^2+^ channel current traces obtained from recombinant HEK-Cav3.1 and Cav3.2 cells, respectively. Traces were recorded at the test potential of 0 mV from the holding potentials of −120 mV. Actions of 25 μM Ni^2+^ (**A**) and 300 μM Ni^2+^ (**B**) on I_Ca.T_ were assessed 5 min after the bath application (green lines) in comparison with control (magenta lines) in the same patch. (**C**) Dose–response curve for the block of I_Ca.T_ by Ni^2+^. Data represent the average (±S.D.) responses from 8 to 12 cells. The smooth curves represent the fit of the Hill equation to the data with IC_50_ of 5.7 μM (Cav3.2, in yellow circles) and 171 μM (Cav3.1, in green circles).

**Figure 2 membranes-12-00686-f002:**
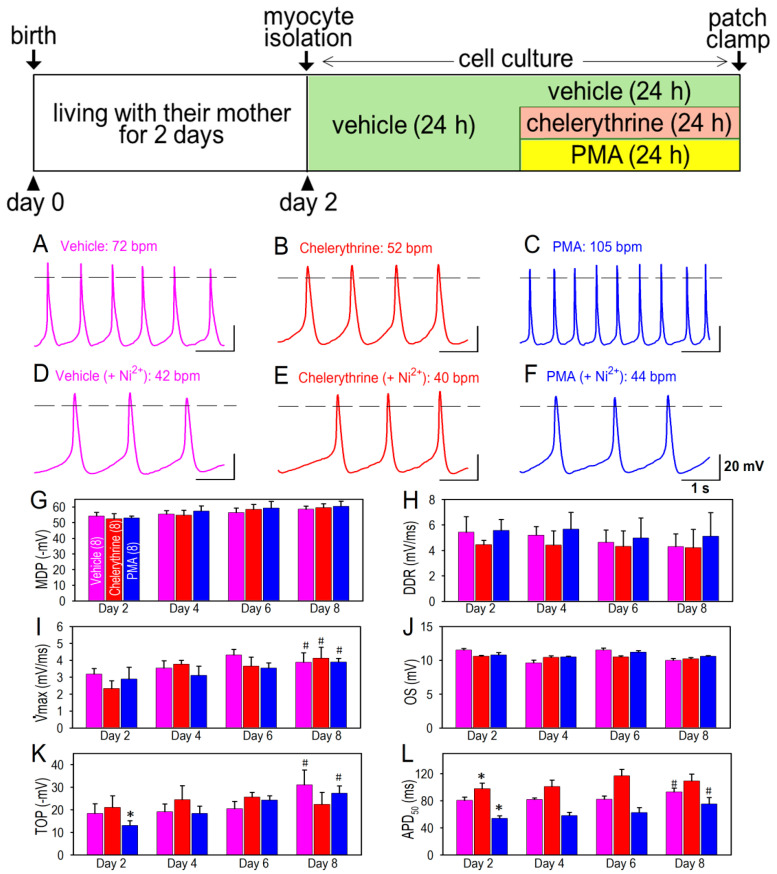
Examples of action potentials and changes of action potential parameters in cardiomyocytes isolated from neonatal rat heart. Representative action potentials recorded in cardiomyocytes at day 2 treated in vehicle (**A**) and in vehicle with 50 μM Ni^2+^ (**D**); treated with a PKC inhibitor chelerythrine (2 μM) (**B**), and chelerythrine (2 μM) with 50 μM Ni^2+^ (**E**); and treated with a PKC activator PMA (1 μM) (**C**), and PMA (1 μM) with 50 μM Ni^2+^ (**F**). Actions of Ni^2+^ were assessed 5 min after application to the bath solution. Note that chelerythrine and PMA were washed out in the culture medium prior to the electrophysiological study. Beating rate is indicated in each panel. Protocols for myocytes isolation and cell culture are shown in the inset above. Action potential parameters representing maximum diastolic potentials (MDP) (**G**); slow diastolic depolarization rate (DDR) (**H**); maximum dV/dt or maximum value of first-time derivatives (Vmax) (**I**); overshoot of action potential (OS) (**J**); take-potential (TOF) (**K**); and action potential duration at 50% repolarization (APD_50_) (**L**). These parameters were measured from the average of 10 consecutive action potentials at day 2, day 4, day 6 and day 8, according to the protocols for myocytes isolation shown in inset in Figure 3. Data were expressed as mean ± S.D. (*n* = 8). * *p* < 0.05, compared with those in vehicle at the same age. # *p* < 0.05, compared with those in the same culture conditions at day 2.

**Figure 3 membranes-12-00686-f003:**
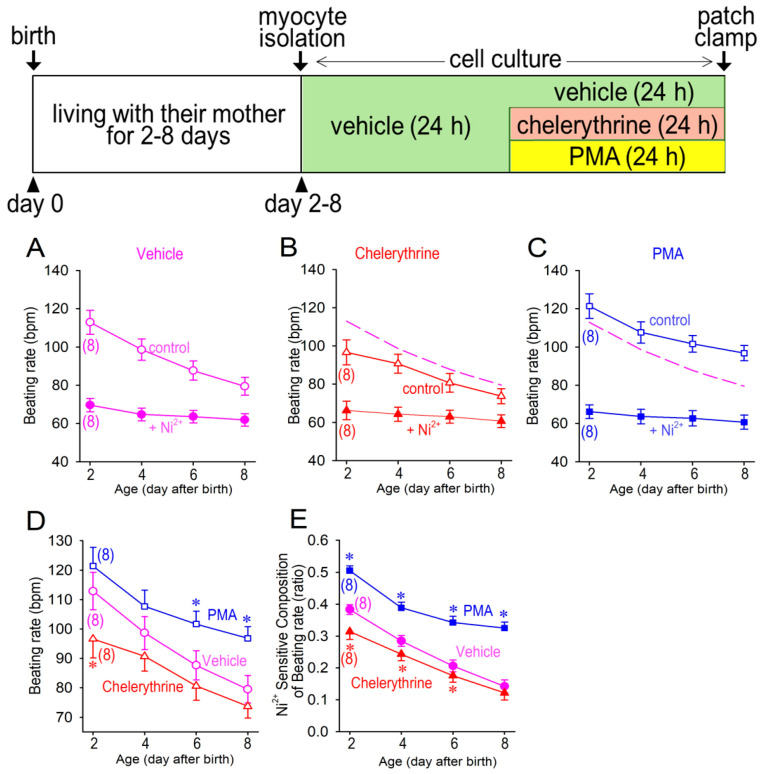
Time-dependent changes of spontaneous beating rates. Myocytes beating rates were plotted against isolation day after birth from day 2 to day 8 treated in vehicle (**A**), with 2 μM chelerythrine (**B**) and 1 μM PMA (**C**). Data with 50 μM Ni^2+^ (open symbols) and without Ni^2+^ application (filled symbols) are shown in each panel. Magenta dashed lines in panels (**B**) and (**C**) represent data without Ni^2+^ (control) in panel (**A**). Changes of beating rates in vehicle, treated with chelerythrine, and PMA without Ni^2+^ application (control) in panels A, B, and C are compared in panel (**D**). Components of Ni^2+^-sensitive beating rates are plotted as a ratio to those without Ni^2+^ (**E**). Actions of Ni^2+^ were assessed 5 min after application to the bath solution. Note that chelerythrine and PMA were washed out in the culture medium prior to the electrophysiological study. Data were expressed as mean ± S.D. (*n* = 8). * *p* < 0.05, compared with those in vehicle at the same age. Protocols for myocytes isolation and cell culture are shown in the inset above.

**Figure 4 membranes-12-00686-f004:**
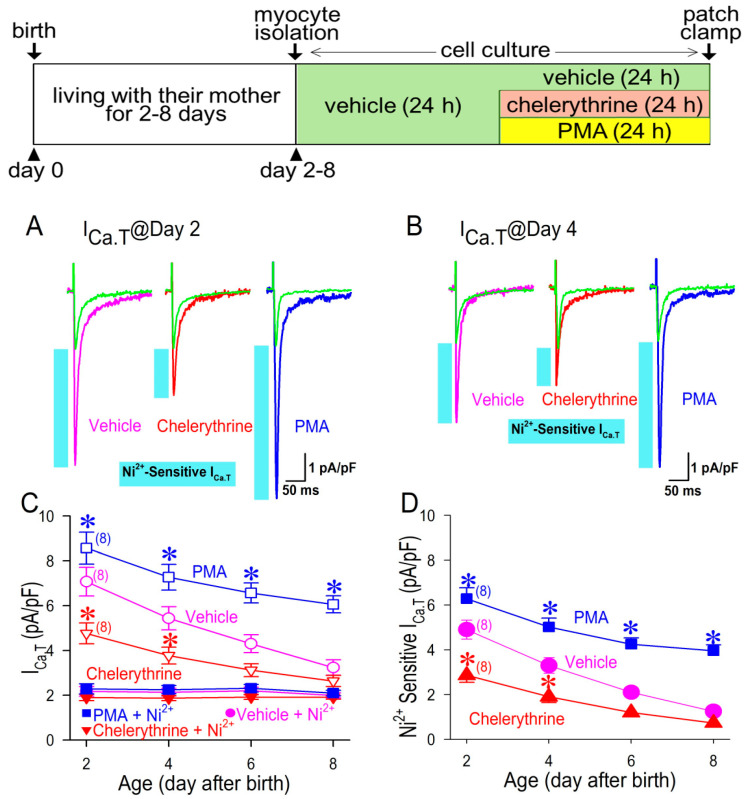
Time-dependent changes of I_Ca.T_. Representative I_Ca.T_ with or without 50 μM Ni^2+^ in myocytes isolated at day 2 (**A**) and day 4 (**B**) after birth, treated in vehicle, 2 μM chelerythrine, or 1 μM PMA. I_Ca.T_ was recorded at the test potential of 0 mV with or without Ni^2+^ application for 5 min in the same patch. Blue bars represent a current component caused by Ni^2+^ or Ni^2+^-sensitive I_Ca.T_. (**C**) Time-dependent changes of I_Ca.T_ with or without Ni^2+^ were plotted against day after birth from day 2 to day 8 treated in vehicle, 2 μM chelerythrine, or 1 μM PMA. (**D**) Time-dependent changes of Ni^2+^-sensitive I_Ca.T_. Note that chelerythrine and PMA were washed out in the culture medium prior to the electrophysiological study. Data were expressed as mean ± S.D. (*n* = 8). * *p* < 0.05, compared with those in vehicle at the same age. Protocols for myocytes isolation and cell culture are shown in the inset above.

**Figure 5 membranes-12-00686-f005:**
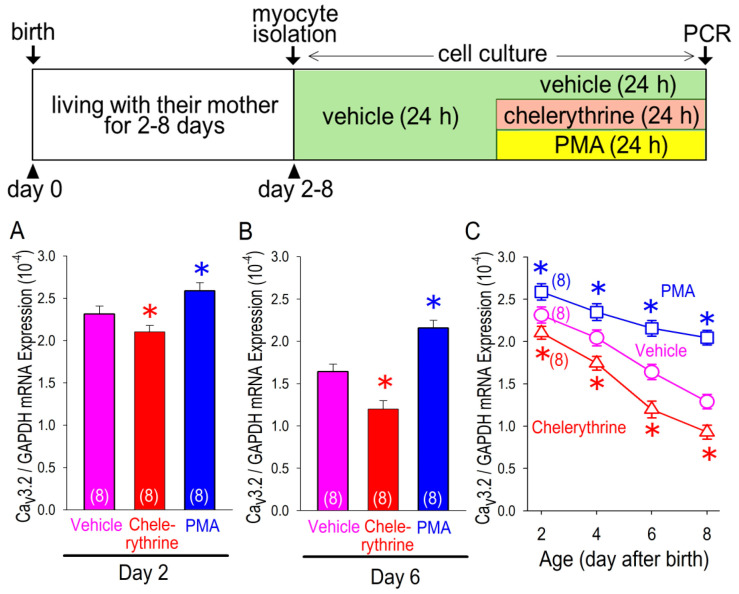
Changes of the levels of Cav3.2-mRNA expression in myocytes. Expression of Cav3.2 mRNA was assessed by real-time PCR in myocytes isolated at day 2 (**A**) and day 6 (**B**). Time-dependent changes of Cav3.2 mRNA were plotted against isolation day after birth from day 2 to day 8 treated in vehicle, 2 μM chelerythrine or 1 μM PMA (**C**). Note that chelerythrine and PMA were washed out in the culture medium prior to the electrophysiological study. Data were expressed as mean ± S.D. (*n* = 8). * *p* < 0.05, compared with those in vehicle at the same age. Protocols for myocytes isolation and cell culture are shown in the inset above.

**Figure 6 membranes-12-00686-f006:**
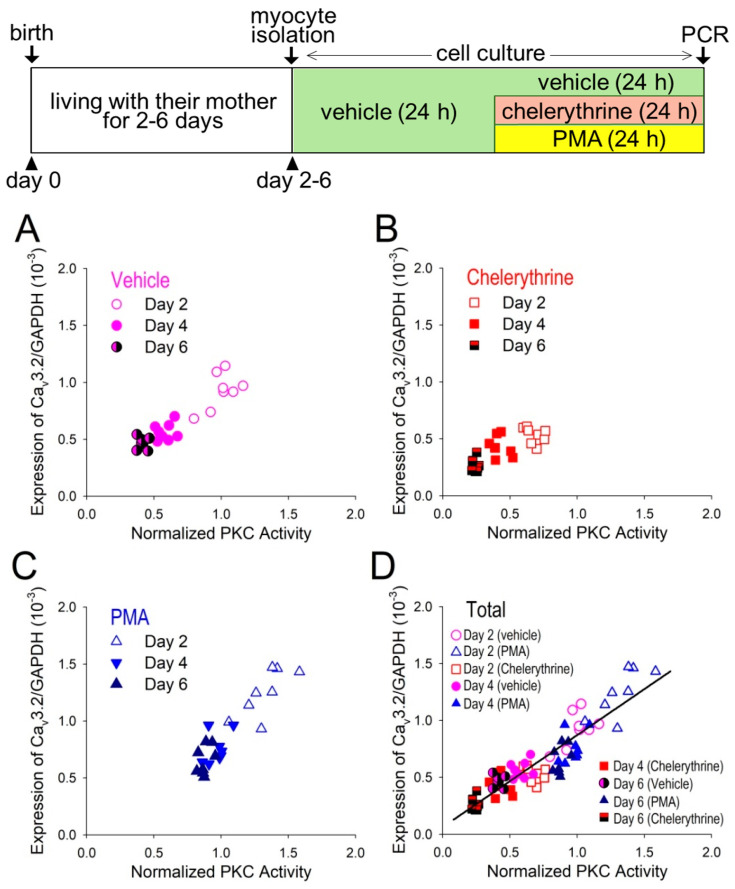
Relation between PKC activity and Cav3.2-mRNA expression. Expression of Cav3.2 mRNA was assessed by real-time PCR and plotted against PKC activity in myocytes isolated at day 2, 4 and 6, treated in vehicle (**A**), 2 μM chelerythrine (**B**), or 1 μM PMA (**C**). Whole data set from myocytes treated in vehicle, chelerythrine or PMA were plotted in terms of Cav3.2 mRNA level against PKC activity and assessed by linear regression analysis with the coefficient of determination (Rsqr) of 0.828 (**D**). Note that chelerythrine and PMA were washed out in the culture medium prior to the electrophysiological study. Protocols for myocytes isolation and cell culture are shown in the inset above.

**Figure 7 membranes-12-00686-f007:**
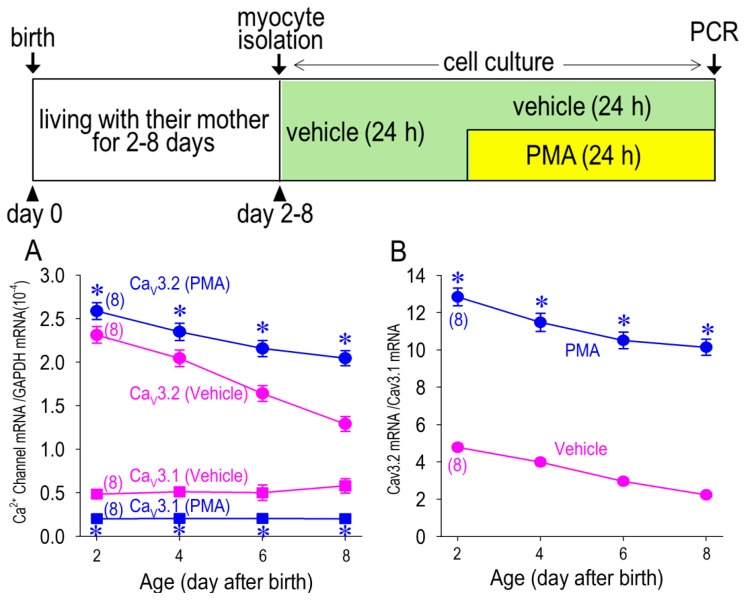
Changes of the levels of Cav3.1 and Cav3.2-mRNA expression in myocytes. Expression of Cav3.1 and Cav3.2 mRNA was assessed by real-time PCR in myocytes isolated at day 2, 4, 6 and 8, treated in vehicle or 1 μM PMA (**A**). The same data for Cav3.2 mRNA from Figure 5C replotted. Time-dependent changes of Cav3.2 mRNA/Cav3.1 mRNA were assessed in cells treated in vehicle or 1 μM PMA (**B**). Note that PMA was washed out in the culture medium prior to the electrophysiological study. Data were expressed as mean ± S.D. (*n* = 8). * *p* < 0.05, compared with those in vehicle at the same age. Protocols for myocytes isolation and cell culture are shown in the inset above. Numbers of experiments are given in parentheses.

**Figure 8 membranes-12-00686-f008:**
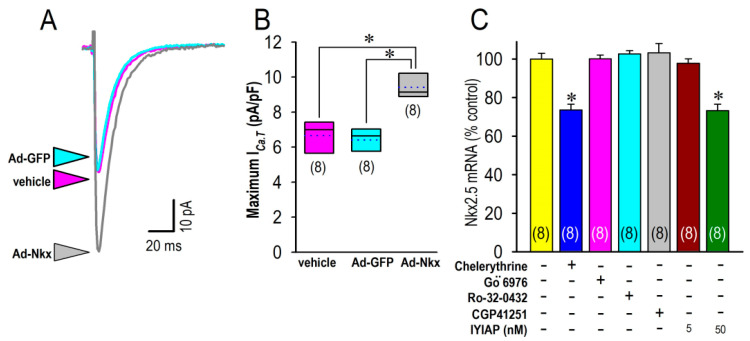
Actions of a transcription factor, Nkx2.5, on I_Ca.T_ associated with PKC activity. (**A**) Representative current traces of I_Ca.T_ at 0 mV from myocyte treated in vehicle (magenta arrowhead), Ad-GFP-treated (blue arrowhead) or Ad-Nkx treated myocytes (gray arrowhead) 24 h after infection of each recombinant adenovirus. (**B**) Group data for the maximum I_Ca.T_ obtained at −20 mV in myocytes treated in vehicle (magenta), Ad-GFP (blue), or Ad-Nkx (gray). Data were expressed as mean (solid line) ± S.D. (height of the box) with the median (dotted line). (**C**) Expression of Nkx2.5 mRNA assessed by real-time PCR in myocytes isolated at day 2 treated in vehicle (yellow); 2 μM chelerythrine (navy); a PKCα inhibitor Gö 6976 of 20 nM (magenta); a PKCα inhibitor Ro-32-0432 of 30 nM (blue); a PKCβI inhibitor CGP41251 of 2 μM (gray); a PKCβI inhibitor IYIAP at 5 nM (brown); or a PKCβII inhibitor of IYIAP at 50 nM (green) applied for 24 h. Amount of each mRNA was normalized to the mean in vehicle (yellow) as assigned as 100. Data were expressed as mean ± S.E. (*n* = 8). * *p* < 0.05, compared with those in vehicle group (yellow). Numbers of experiments are given in parentheses.

## Data Availability

The data in this study are available from the corresponding author on reasonable request.

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
