# Peer review of "Protein Kinase C Regulates Expression and Function of the Cav3.2 T-Type Ca2+ Channel during Maturation of Neonatal Rat Cardiomyocyte"

_membranes, 2022, doi:10.3390/membranes12070686_

Round 1

Reviewer 1 Report

The article is very interesting, as the main aim of the study is to identify the molecular mechanism that governs the isoform switching of the T-type Ca channels from the Cav3.2 channel to the Cav3.1 channel in the cardiomyocytes during the perinatal phase using the whole-cell patch-clamp technique and mRNA quantification.

General comments:

  1. The abstract and the introduction are clearly define the objectives and main conclusion.
  2. In the Materials and Method section
    • why the authors performed the whole-cell patch clamp technique experiments at 37ºC, normally the experiments are performed at 20ºC
    • The used of the dimethyl sulfoxide (DMSO), even in very low concentrations should be avoided. I never use DMSO in patch clamp technique because I always have interference with the ionic currents. You must demonstrate a control with DMSO or dissolve the inhibitor in ethanol.
    • It should be added how the rundown of calcium currents was eliminated, what mathematical formula was used. this information should be added to the Data acquisition and statistical analysis section.
    • I also did not see the number of cells used in the experiments and the average cell size in pF. Were these pF very variable? this information should be added.
    • Add the statistical test used in the figure legend.
  3. The results and conclusion are good.

Author Response

To reviewer #1

Thank you for your very careful reading of our manuscript, and your favorable suggestions on our study.  In this revision, we have tried to address each point, and we feel that the manuscript has been improved because of your valuable comments.  We hope that these changes in manuscript are satisfactory and that the revised version is acceptable for publication in Membranes.  Changes/additional text are yellow-highlighted for clarity. 

  1. Why the authors performed the whole-cell patch clamp technique experiments at 37ºC, normally the experiments are performed at 20ºC.

Thank you for your question on the patch clamp study. As you have pointed out, it is quite common for many electrophysiologists to perform patch clamp study at room temperature or at the temperature around 20oC. Actually we have lots of patch clamp studies performed at room temperature, because relative low temperatures lead to slow activation and slow inactivation gating, which is suitable for accurate analysis of the channel kinetics. However, in this study, we performed patch clamp study at the condition of 37oC. This is because of the following reasons; 1) Neonatal cardiomyocytes show very slow or sometimes very irregular spontaneous beating at room temperatures. To observe and evaluates beating rates at each condition, higher bath temperature was required. Also because current clamp experiments (Figure 2, 3) and voltage clamp experiments (Figure 1, 4, 8) need to be performed at the similar conditions including bath temperature, we performed all patch clamp studies at 37oC. 2) The chamber heating system in this study (Bipolar Temperature Controller, model TC-202A, Harvard Apparatus, Holliston, MA, USA) did not make any bad electrical noise, which was very favorable for patch clamp experiments. 

  1. The used of the dimethyl sulfoxide (DMSO), even in very low concentrations should be avoided. I never use DMSO in patch clamp technique because I always have interference with the ionic currents. You must demonstrate a control with DMSO or dissolve the inhibitor in ethanol.

I perfectly agree with you on this point. I do not like DMSO as a vehicle in the patch clamp study because we also know DMSO has non-specific effects on ion channel currents as a short-term effect. Thus I always force my group members not to use DSMO in the stock solution. Particularly in addition, we do know that DMSO has very annoying actions for cell growth and differentiation; DMSO has an impact on the epigenetic regulatory system, changes the genome-wide DNA methylation status, and induces formation of electrophysiological alterations as we have previously reported (J Physiol Sci 70:39, 2020).  However, when experimental chemicals are not solved in ethanol, there is no better alternative other than DMSO, unfortunately.  This was the case we unwillingly applied DMSO for the stock solution of Ro-32-0432, Gö 6976 and IYIAP. Please accept that we used DMSO as a last resort in this study, and we would like to ask for your understanding of this. Control studies with DMSO are shown as Vehicle in the text and figures.

  1. It should be added how the rundown of calcium currents was eliminated.

Thank you for your specific comment on the Ca2+ channel characteristics. As you have pointed out, the activity of the L-type Ca2+ channels decreases when the cytoplasmic side of the channels is perfused with an artificial physiological solution in whole-cell recording, namely “run-down”. However, T-type Ca2+ channels currents are found to be resistant to rundown (Physiol Rev. 2003 83:117-161). For the recording of ICa.T in Figure 4, ICa.L was initially recorded at the beginning of the study for the current subtraction.  This current trace was applied before and after the application of Ni2+ in the same patch. Thus we believe that rundown of L-type Ca2+ channel current, if any, does not affect the measurement of ICa.T in this experimental protocol

  1. What mathematical formula was used. this information should be added to the Data acquisition and statistical analysis section.

In response to your request, Hill plot equation is added to the Methods section in this revision.

  1. I also did not see the number of cells used in the experiments and the average cell size in pF. Were these pF very variable? this information should be added.

We are sorry that cell numbers in the experiments in Figure 1 were only described in the caption In figure 1, each symbol represents the average from 8-12 cells. Because yellow circles (Cav3.2) for 25 μM and 30 μM are nearly overlapping, and ones for 250 μM and 300 μM are nearly overlapping, it is not evident to put cell number parentheses nearby the symbols in panel C in Figure 1. Same for the green circles (Cav3.1) for 25 μM and 30 μM, as well as 250 μM and 300 μM.  Because of these graphical reasons, we have just described the cell numbers only in the caption.  We would like to ask for your understanding.  Cell numbers are shown in each panel in experiments in figures 3-5, and 7-8. Cav3.1- and Cav3.2-HEK cells utilized in the heterologous expression system in figure 1 were nearly perfectly isoform in terms of cell capacitance (15pF-16pF). Thus we applied the net ICa.T amplitude to assess the inhibitory actions of Ni2+ in figure 1. Cell capacitance of the neonatal cardiomyocyte was various depending on the age (days). Although the cell capacitance tended to be larger in cardiomyocytes at age 8 (22.9 ± 3.6 pF in vehicle) in comparison with those at age 2 (16.7 ± 2.5 pF in vehicle), there was no significant difference among the groups regardless of the presence or absence of PKC inhibitor/activator.  This description was added to the text in the revision.

Reviewer 2 Report

Wang and co-workers examined the effects of pharmacological PKC modulators on changes in electrical automaticity, T-type current amplitude, and Cav3.1 / Cav3.2 mRNA expression in postnatal day 2-8 rat ventricular myocytes. They found that 24-h PKC activation and inhibition increased and decreased, respectively, automaticity and that changes in automaticity were accompanied by proportional changes in the Ni2+-sensitive T-type current (encoded by Cav3.2), but not Ni2+-insensitive (encoded by Cav3.1) T-type current component. PKC-mediated overexpression of the transcription factor Nkx2.5 caused an increase in T-type calcium current in ventricular myocytes. These findings lead the authors to conclude that changes in myocyte automaticity during early postnatal development are mediated by activation of a PKC-Nkx2.5-Cav3.2 signaling pathway in rat ventricular cardiomyocytes.

Major comments:

  • T-type currents were elicited at a single potential only, i.e., 0 mV. Since maximal amplitudes of peak T-type currents evoked in ventricular cardiomyocytes at low millimolar external calcium concentrations typically occur at negative potentials (e.g., Physiol Res 53:369-377, 2004), the rationale for using 0 mV is not clear. Also, L-type currents typically peak around 0 mV under these ionic conditions, complicating a separation of the two current types. The authors should measure the whole-cell peak current density-voltage relations and re-examine the use of 0 mV in the experimental protocols.
  • Changes in the voltage-dependence of steady – state activation and inactivation as well as those in recovery from inactivation can contribute to the time-sensitive changes in peak T-type current seen in neonatal cardiomyocytes. The authors should examine whether modulation of PKC activity, in addition to changing T-type current density, also alters biophysical properties of T-type currents.
  • While Ni2+ at a concentration of 50 microM appears to selectively inhibit Cav3.2 T-type currents in an exogenous expression system (Figure 1), its selectivity for inhibiting T-type channels in native rat ventricular cardiomyocytes is not documented. Since it is known that Ni2+ inhibits both L-type channels and the sarcolemmal Na/Ca exchanger at micromolar concentrations in mammalian cardiomyocytes, and both conductances contribute to cell automaticity, information about Ni2+ effects on L-type current and Na/Ca exchange current should be provided.
  • Examples of action potentials shown in Figure 2 suggest that PKC activity regulates spontaneous beating rate by modulating the rate of slow depolarization and/or shifting the take-off potential. Comprehensive analyses of the action potential parameters (maximal diastolic potential, rates of slow and fast depolarization, take-off potential, peak amplitude and action potential durations) should be provided. Please add a voltage scale to panels A through F of Figure 2.
  • It is shown that pharmacological PKC activation with PMA enhances Cav3.2 gene transcription which correlates with enhanced peak T-type currents in the myocytes. The authors should also perform the control experiment using the inactive 4-alphaPMA. Also, they should perform immunocytological studies to examine whether the PMA-induced increase in Cav3.2 gene transcription correlates with increased surface expression of the channel.
  • 5 overexpression leads to an increase in peak T-type calcium current (Figure 8) which is inhibited by chelerythrine and IYIAP. Was increased Nkx2.5 expression accompanied by increased beating rate of the cardiomyocytes?

Author Response

To Reviewer #2

              Thank you for your careful reading of our manuscript, and your very important questions, which need to be clarified.  In this revision, we have tried to address each point, and we feel that the manuscript has been improved because of your valuable comments.  We hope that these changes in manuscript are satisfactory and that the revised version is acceptable for publication in Membranes.  Changes/additional text are yellow-highlighted for clarity. 

  1. T-type currents were elicited at a single potential only, i.e., 0 mV. Since maximal amplitudes of peak T-type currents evoked in ventricular cardiomyocytes at low millimolar external calcium concentrations typically occur at negative potentials (e.g., Physiol Res 53:369-377, 2004), the rationale for using 0 mV is not clear. Also, L-type currents typically peak around 0 mV under these ionic conditions, complicating a separation of the two current types. The authors should measure the whole-cell peak current density-voltage relations and re-examine the use of 0 mV in the experimental protocols.

Thank you for your specific comment on the voltage-gated Ca2+ channel gating. As an electrophysiological laboratory, we have experienced T-type Ca2+ channel investigations for more than 20 years. Nevertheless, we are not confident that we precisely measure ICa.T in this experimental protocol in neonatal cardiomyocyte; our ICa.T recordings in Figure 4 potentially include Cav3.1-ICa.T, Cav3.2-ICa.T, Cav1.2-ICa.L, Cav1.3-ICa.L and powerful Nav1.5-INa to a certain extent.  Therefore, electrophysiological study cannot completely discriminate Cav3.2-ICa.T even with the use of Ni2+ and 30 μM TTX in the bath solution. Therefore we believe that current clamp and voltage clamp experiments only qualitatively suggest the isoform changes of the T-type Ca2+ channel in response to PKC activity.  Thus we applied other experimental technique in experiments in figure 5-8 to make the conclusion. The rationale for using 0 mV as the test potential is; 1) We needed to minimize the interference of Cav1.3-ICa.L on the T-type Ca2+ channel current, because Cav1.3-ICa.L can be activated even more hyperpolarized potentials than those of ICa.T, 2) The conductance of the T-type Ca2+ channel reaches a maximum value at the membrane potentials of -20 mV or more depolarized potentials, which is important to minimize the surface charge effect of Ni2+ as a divalent cation on the measurement of ICa, 3) We needed to lessen the impact of the time-dependent shift of activation curve towards negative direction, although the shift of ICa.T was not as dominant as that of INa. Because of these reasons, we believe that evaluation of ICa.T at the test potential of 0 mV was reasonably good enough.

  1. Changes in the voltage-dependence of steady – state activation and inactivation as well as those in recovery from inactivation can contribute to the time-sensitive changes in peak T-type current seen in neonatal cardiomyocytes. The authors should examine whether modulation of PKC activity, in addition to changing T-type current density, also alters biophysical properties of T-type currents. While Ni2+ at a concentration of 50 microM appears to selectively inhibit Cav3.2 T-type currents in an exogenous expression system (Figure 1), its selectivity for inhibiting T-type channels in native rat ventricular cardiomyocytes is not documented. Since it is known that Ni2+ inhibits both L-type channels and the sarcolemmal Na/Ca exchanger at micromolar concentrations in mammalian cardiomyocytes, and both conductances contribute to cell automaticity, information about Ni2+ effects on L-type current and Na/Ca exchange current should be provided.

Thank you for your comments. As an electrophysiologist of cardiac ion channels, I am interested in these biophysical features of the channel.  However, this investigation was purposed to explore the signals that governs the changes of isoforms in the T-type Ca2+ channels.  For this purpose, I would like to confess, biophysical analysis of the channel gating is less helpful and can provide only limited information.  Particularly, the potential presence of the Cav1.3 channel may cause inconvenience on the further analysis of ICa.T gating in neonatal cardiomyocytes, because we do not have any tool to completely separate ICa.T from Cav1.3-ICa.L in terms of selective inhibitors or pulse protocols. As you have pointed out, Ni2+ can inhibit multiple channels and NCX as well. Albeit we do understand the lack of electrophysiological evaluation in this study, we also comprehend that additional biophysical analysis of the T-type Ca2+ channel may not provide further conclusion regarding the function of a transcription factor Nkx2.5 and specific PKC isoforms.  I would like to ask for your understanding of the limitation of patch clamp study in this context. 

  1. Examples of action potentials shown in Figure 2 suggest that PKC activity regulates spontaneous beating rate by modulating the rate of slow depolarization and/or shifting the take-off potential. Comprehensive analyses of the action potential parameters (maximal diastolic potential, rates of slow and fast depolarization, take-off potential, peak amplitude and action potential durations) should be provided.

In response to your request, we newly reanalyzed our action potentials recordings, and put them in Figure 2 in this revision. Probably because neonatal cardiomyocytes in this study were nearly all derived from ventricular cardiomyocytes, spontaneous beatings were relatively irregular (figure 2 A-F). Although the diastolic depolarization rate (DDR) tended to be decreased after the birth, we were unable to observe the significant change during the observation period up to 8 days with chelerythrine or PMA. This is probably because the beating rates were decided in a complex way in terms of the balance with the maximum diastolic potentials, DDR, take-off potentials, voltage ranges of the window current of ICa.L and ICa.T, density of INa, activation voltage of INa, and so forth. In this context, we do understand the limitation of electrophysiological method to determine the quantitative contribution of Cav3.1- and Cav3.2-T-type Ca2+ channel to action potentials in neonatal cardiomyocytes.

  1. Please add a voltage scale to panels A through F of Figure 2.

We are sorry for this. Voltage scales are now shown in the revision.

  1. It is shown that pharmacological PKC activation with PMA enhances Cav3.2 gene transcription which correlates with enhanced peak T-type currents in the myocytes. The authors should also perform the control experiment using the inactive 4-alphaPMA. Also, they should perform immunocytological studies to examine whether the PMA-induced increase in Cav3.2 gene transcription correlates with increased surface expression of the channel.

We agree with you that these additional experiments will make the study perfect regarding the positive actions of PKC for Cav3.2 transcription. However, these additional experiments may need a few months perform and we are unable to revise the manuscript within a couple of weeks. We would like to ask for your understanding.

  1. overexpression leads to an increase in peak T-type calcium current (Figure 8) which is inhibited by chelerythrine and IYIAP. Was increased Nkx2.5 expression accompanied by increased beating rate of the cardiomyocytes?

Thank you for your interest on a transcription factor NKx2.5.  Since we have previously demonstrated an increase of spontaneous beating by overexpression of Nkx2.5 in neonatal cardiomyocytes (Reference #18 J Mol Cell Cardiol 2007, 42, 1045-1053 ), we did not repeat the similar experiments in this study. In this revision we have added these information to the text in regard to you comment.

Reviewer 3 Report

In this manuscript by Wang et al the authors describe their work investigating the role of protein kinase C  in regulation of T-type Ca2+ channel, Cav3.2, during maturation of rat cardiomyocytes.  The presented results demonstrate that activation of PKC by PMA  upregulates Ni2+-sensitive T-type Ca2+ current and accelerates Ni2+-sensitive beating rate and in neonatal cardiomyocytes. Whereas PKC inhibition has an opposite effect. Strong positive correlation between Cav3.2 -mRNA expression and PKC activity, as well as higher sensitivity of Cav3.2 to Ni2+ compared to Cav3.1, suggest that PKC dependent component of the T-type Ca2+ current in embryonic cardiomyocytes is mediated by Cav3.2 channels. I do not have any major criticisms or this manuscript, however, some minor points below need to be addressed:

1.   Figure 1, panel C. Concentration on the X axis should be in micromolar.

2. Line 241. “Other action parameter parameters…” – action potential

3. Text following line 246 appears in the middle of the Figure 3. There are also some text lines in the middle of Figure 7.

4. Line 598. “Actions of PKC for the Cac3.2 channel” – Cav3.2

5. Line 604. “Fatal” – foetal

Round 2

Reviewer 1 Report

The authors have revised the manuscript according to the reviewer's comments.

I recommend accept

Reviewer 2 Report

In the revised version of their manuscript, the authors provide quantitative analyses of action potential parameters, but otherwise have not performed additional experiments as suggested. Thus, most of the concerns remain.

Specifically,

-          The Ni2+-sensitivity and voltage-dependent gating properties of native T-type channels have not been determined. However, such measurements are critical for interpretation of the experimental findings. For example, the authors report maximal diastolic potentials ranging from ~-55 mV to ~-60 mV. Their voltage-clamp protocol uses a holding potential of -50 mV to suppress T-type channels. If T-type channels are completely inactivated at a holding potential of -50 mV, then one would expect these channels to only minimally, if at all, contribute to the whole-cell current in neonatal cardiomyocytes, unless the authors provide direct experimental evidence to the contrary.

-          Traces shown in Figure 2 are not representative because the beating rates in Figure 2 are markedly off their respective mean values shown in Figure 3.

-          If the authors do not find significant differences in the beating rate-determining factors, i.e., DDR, MDP, TOP, APD, following treatment with a PKC blocker or activator, then how do  they explain the significant changes in beating rate occurring with each pharmacological intervention (Figure 3) or those occurring over time (Figure 3)?

-           Analyses of the effects of Ni2+ on action potential parameters are not provided.

-          Although the effect of NKX2.5 overexpression on T-type calcium current was assessed, its modulation of action potential properties was not.

Overall, evidence for role of changes in Cav3.2 expression in regulating automaticity of postnatal rat cardiomyocytes remains circumstantial, at the most.

This manuscript is a resubmission of an earlier submission. The following is a list of the peer review reports and author responses from that submission.

Round 1

Reviewer 1 Report

In their study entitled “Protein Kinase C Regulates Expression and Function of The Cav3.2 T-type Ca2+ Channel during Maturation of Neonatal Rat Cardiomyocyte”, Wang et al present data suggesting that PKC activity could control Cav3.2 channel expression through the transcription factor Nkx2.5. While of interest, this study presents too many issues needing to be addressed before this manuscript becomes acceptable for publication.

Major comments:

The use of nickel as a discriminant between Cav channels is understandable, but doses used in HEK overexpression models lacking the accessory subunits may not reflect the inhibition taking place in cardiomyocytes where the subunits are expressed.

In their experiments, the authors use very high concentrations of PMA (1 mM instead of the reported 100 nM usually found in literature) and chelerythrine (2 mM instead of 5-10 µM). What is the reason for using such high concentrations? Can the results reported by the authors be linked solely to PKC modulation, or can these high concentrations elicit non-specific effects, especially when cells were generally treated 24h with these drugs?

Lines 255-266: the authors conclude from the observed effects of PKC modulators on cardiomyocytes that Cav3.2 expression is controlled by PKC. The authors however never consider that PKC, as previously reported, can directly phosphorylate this channel and increase its activity independently of any effect on its expression. Moreover, as acknowledged by the authors in the discussion section, no Western blot, biotinylation or immunofluorescence results are presented to support their claim that channel expression is indeed significantly increased in PMA treated cells compared to control or chelerythrine conditions. Without these key results, the authors’ claim that PKC controls Cav3.2 persistence in cardiomyocytes appears premature.

Figure 5B: the values presented in this figure are not consistent with those shown in figure 5C. Is legend supposed to say “day 8”?

Figure 6A: how do the authors explain the apparent decrease in PKC activity observed in control condition? Is PKC expression (mRNA, protein) decreasing over time? Why was day 8 omitted for this experiment? Can the decrease in Cav3.2 activity/expression be correlated to PKC decrease in activity if the latter is not significantly altered between days 4, 6 and 8 (a result potentially found in all 3 experimental conditions presented in figure 6)?

The possible involvement of Nkx2.5 in Cav3.2 regulation by PKC is an interesting idea. Several key controls are however missing in the present manuscript: the authors do not show that overexpression of Nkx2.5 in their model indeed increases Cav3.2 expression (mRNA, protein); no siRNA/shRNA against Nkx2.5 are proposed to validate its impact on Cav3.2 expression and activity; no experimental confirmation (50 µM Ni2+) that the current shown in figure 8A is exclusively due to Cav3.2 is proposed by the authors; PMA effect on Nkx2.5 expression is not shown.

Minor comments:

Scale on figure 1C is wrong: nickel concentration should be expressed in µM.

There is no scale on figures 4A and 4B.

Author Response

Manuscript ID: membranes-1643535; Response to Reviewer #1

       Thank you for your very careful reading of our manuscript, and your favorable suggestions on our study.  In this revision, we have tried to address each point, and we feel that the manuscript has been improved because of your valuable comments.  We hope that these changes in manuscript are satisfactory and that the revised version is acceptable for publication in the special issue “Physiology, Pathophysiology and Pharmacology of Calcium Channels” in Membranes.  Changes/additional text are yellow-highlighted for clarity. 

Major comments:

  1. The use of nickel as a discriminant between Cav channels is understandable, but doses used in HEK overexpression models lacking the accessory subunits may not reflect the inhibition taking place in cardiomyocytes where the subunits are expressed.

       Thank you for your expertise question on the voltage-gated Ca2+ channel. However, as far as we know, T-type Ca2+ channels consist on a single Cav3 pore forming α1 subunit (α subunit) that contains all the structural determinants of the channel gating and ion selectivity and permeability unlike other voltage-gated Ca2+ channels. Thus no auxiliary subunits (β, γ, α2/δ) are required for the normal function of the T-type Ca2+ channel. With this reason, we believe that IC50 values of Ni2+ for the inhibition Cav3 from the heterologous expression system in HEK293 cells shown in Figure 1 may represent physiological actions of Ni2+ to block ICa.T in cardiomyocytes.

In their experiments, the authors use very high concentrations of PMA (1 mM instead of the reported 100 nM usually found in literature) and chelerythrine (2 mM instead of 5-10 µM). What is the reason for using such high concentrations? Can the results reported by the authors be linked solely to PKC modulation, or can these high concentrations elicit non-specific effects, especially when cells were generally treated 24h with these drugs?

       We are very sorry for these bad mistakes. Concentration for PMA was 1 μM but not 1 mM. Concentration for chelerythrine was 2 μM but not 2 mM. The original draft was made in Time New Roman font. When the manuscript was converted to the MDPI format in font of Palatine Linotype, some “μM” was incorrectly converted to “mM” by unknown reason, which we did not notice in a careless manner. Again we apologize for these very bad mistakes. Description on these concentration units have been double-checked and converted to “μM” in this revision.

Lines 255-266: the authors conclude from the observed effects of PKC modulators on cardiomyocytes that Cav3.2 expression is controlled by PKC. The authors however never consider that PKC, as previously reported, can directly phosphorylate this channel and increase its activity independently of any effect on its expression. Moreover, as acknowledged by the authors in the discussion section, no Western blot, biotinylation or immunofluorescence results are presented to support their claim that channel expression is indeed significantly increased in PMA treated cells compared to control or chelerythrine conditions. Without these key results, the authors’ claim that PKC controls Cav3.2 persistence in cardiomyocytes appears premature.

       First of all, we apologize for the poor description on the patch clamp methods. For the recording of action potentials and ICa.T, PMA or chelerythrine was not included in the bath solution.  PMA and chelerythrine were included in the culture medium for 24 hours, and they were washed out prior to the electrophysiological study. Thus we believe that acute actions of PKC for the channel phosphorylation could be minimized in this study. To clarify the methods for electrophysiology, we add some description in the methods section and figure legends in this revision. Nevertheless, as you pointed out, we cannot definitely conclude that PKC regulates Cav3.2 expression.  Because this study is largely dependent on the electrophysiological results, interpretation of the result should be different from those made by molecular biological studies. Molecular biological examination focusing on the Nkx2.5 and Cav3.2 would be necessary to support the conclusion.  Actually we would like to continue the study focusing on Nkx2.5 in the heart with respect to PKC isoform actions. Although we cannot correspond to your question/comment based on new experiments, we would like to ask for your kind understanding for the present result based on the patch clamp study, because of the limited revising period for 10 days. We have changed the tone of conclusion in the manuscript, and would like to accept any suggestions you may make on the wordings. 

Figure 5B: the values presented in this figure are not consistent with those shown in figure 5C. Is legend supposed to say “day 8”?

       We are sorry for this mistake. Figure 5 Panel B represents changes in Day 6.  This mistake has been corrected in the revision.

Figure 6A: how do the authors explain the apparent decrease in PKC activity observed in control condition?

       We do not know the mechanism for the declining of PKC activity after the birth in the control condition.  The mechanism for this may be involved in the maternal factors including gonadotropins and sex hormones derived from maternal serum circulation to the new-born rat, although we do not have data to support this. Because we do not have data, this speculation is only described as a postulation in the discussion section in the last paragraph of 4.1.

Is PKC expression (mRNA, protein) decreasing over time?

       We did not examine PKC-proteins because we have no experience to assay these molecules in active form. PKC activation is one thing, expression of PKC proteins are probably another.  We do not have any experience to assay PKC activities other than the enzyme-linked immunosorbent assay (ELISA).  Also because our laboratory studies ion channel function with a major focus on electrophysiology, the experimental tools we are using are based on patch clamp methods and some related ones. Please understand the limitation of the method we can apply in this study.

Why was day 8 omitted for this experiment?

       We measured PKC activities by use of the PKC Kinase Activity Kit (ELISA) (EKS-420A; Stressgen Bioreagents Corp., Victoria, British Columbia, Canada). This kit gave us the ability to an end-point or kinetic assay read-out in a convenient 96-well plate.  Although the place is equipped with 96 wells, left two lanes (16 wells) are for blank corrections. Therefore we were able to measure only 80 samples at the best. Data for a single day are composed of three different conditions including vehicle (n=8), PMA (n=8) and chelerythrine (n=8). In the experiment in Figure 6, we examined PKC activities for day 2 (n=24), day 4 (n=24) and day 6 (n=24). So we have used 72 wells for this study. For the evaluation of PKC activities at day 8, 24 additional wells or new kit was necessary. Because assay kit (EKS-420A) was rather expensive, we decided not to evaluate PKC activity on day 8. We would like to ask for your understanding for the financial limitation on this study.

Can the decrease in Cav3.2 activity/expression be correlated to PKC decrease in activity if the latter is not significantly altered between days 4, 6 and 8 (a result potentially found in all 3 experimental conditions presented in figure 6)?

       We are sorry we cannot get your point in this question. Is your question on the Cav3.2/PKC ratio? Is your question on the data point overlap of vehicle day 6 and chelerythrine day 2? Because there is a significant linear relation between PKC activity and expression of Cav3.2 with Rsqr of 0.828 regardless of the difference of days or presence of PKC inhibitor/activator as shown in Figure 6, we made a conclusion that expression of Cav3.2 was highly correlated with PKC activity. Even if the PKC activity was not significantly altered by PMA or chelerythrine, insignificant changes of Cav3.2 could account for the small changes in PKC thereafter. We are sorry for this probable incomplete answer because we are not sure the point.

The possible involvement of Nkx2.5 in Cav3.2 regulation by PKC is an interesting idea. Several key controls are however missing in the present manuscript: the authors do not show that overexpression of Nkx2.5 in their model indeed increases Cav3.2 expression (mRNA, protein);

       Data demonstrating upregulation of Nkx2.5 proteins and Cav3.2-mRNA in neonatal cardiomyocyte by recombinant adenovirus encoding Nkx2.5 (AdNkx2.5) were reported in our previous publication (J Mol Cell Cardiol 2007, 42, 1045-1053). Because the construction of AdNkx2.5 and the way of cell transfection in this study was identical to those in the previous publication, we did not repeat the result in this article again. Some description on the preparation and confirmation of the ability of Ad-Nkx has been added to the method section in the revision.

no siRNA/shRNA against Nkx2.5 are proposed to validate its impact on Cav3.2 expression and activity;

       We agree that experiment with Nkx2.5-siRNA for evaluation of Cav3.2 expression will be powerful to support our conclusion; we wish we had performed the proposed experiment.

no experimental confirmation (50 µM Ni2+) that the current shown in figure 8A is exclusively due to Cav3.2 is proposed by the authors;

       We applied Ni2+ to the recording chamber solution to discriminate Cav3.2-ICa.T as a possible substitute for PKC-dependent component of ICa.T as a starting experiment in this study. By the experiment in Figure 6, we have confirmed the positive correlation between PKC and Cav3.2.  Because AdNkx2.5 has no regulatory mechanism for Cav3.1-ICa.T (J Mol Cell Cardiol 2007, 42, 1045-1053), we feel that electrophysiological experiment by use of Ni2+ on ICa.T in Figure 8 may be redundant in this context.   

PMA effect on Nkx2.5 expression is not shown.

       We partially agree with you; PCR study in Figure 8C was designed to determine the PKC isoforms that were responsible for Nkx2.5 expression. For this purpose, we carefully applied several different chemicals (and concentrations) for the selective inhibition of PKC isoforms.  We thought PMA was not useful suitable for this purpose. However, after you pointed out this, we feel that we should have tried PMA in this experiment as a positive control.

Minor comments:

Scale on figure 1C is wrong: nickel concentration should be expressed in µM.

         We are sorry that scale label of μM was changed to mM when the original figure (constructed by a software SigmaPlot 13) was converted to jpg format. This misspelling has been corrected in the revision.

There is no scale on figures 4A and 4B.

       Scales for time and current amplitude were placed in Figure 4A, B.

Reviewer 2 Report

The authors set out to investigate the functional expression of T-type calcium channels, Cav3.1 and Cav3.2, in the maturation of neonatal rat cardiomyocytes. The authors conclude with the data presented in the manuscript that the abundant expression of Cav3.2, in neonatal cardiomyoctes, and its age dependent decline is attributable to protein kinase C activity via its effects on expression of the cardiac NKx2.5 transcription factor. The study overall is well written and the experimental approaches sound,  however I have some concerns that are addressed below:

The novelty of the findings is difficult to decipher with the current state of the introduction section. The authors have a several papers which have dealt with the maturation switch of T-type calcium channels in maturating neonatal cardiomyocytes and PKC dependency (Zheng et al 2010) as well as the possible role of the cardiac Nkx2.5 transcription factor in regulating Cav3.2 expression (Wang et al 2007). The authors bring up briefly some of these points in the discussion (i.e. short term vs long term effects) however the setting of why these experiments and the novelty of what they would bring is not clearly defined, and only slightly addressed in the discussion.

Have the authors considered whether calcium dependent mechanisms (i.e. activity dependent TFs) are mediating the expressional/current density changes observed? Would clamping intracellular Ca2+ (i.e. with BAPTA-AM) in the presence of PKC activation augment the expressional changes observed?

Overexpression of Nkx2.5 was shown to effect the maximum Icat current after 24h. Where there changes observed in cellular automaticity?

Is it possible that this study could shed light on Cav3.2 associated expressional changes observed in pathological cardiac hypertrophy?  

Author Response

Manuscript ID: membranes-1643535; Response to Reviewer #2

Thank you for your careful reading of our manuscript, and your favorable comments on our study.  In this revision, we have tried to address each point, and we feel that the manuscript has been improved because of your valuable comments.  We hope that these changes in manuscript are satisfactory and that the revised version is acceptable for publication in Membranes.  Changes/additional text are yellow-highlighted for clarity. 

Comments

1.   The novelty of the findings is difficult to decipher with the current state of the introduction section. The authors have a several papers which have dealt with the maturation switch of T-type calcium channels in maturating neonatal cardiomyocytes and PKC dependency (Zheng et al 2010) as well as the possible role of the cardiac Nkx2.5 transcription factor in regulating Cav3.2 expression (Wang et al 2007). The authors bring up briefly some of these points in the discussion (i.e. short term vs long term effects) however the setting of why these experiments and the novelty of what they would bring is not clearly defined, and only slightly addressed in the discussion.  Have the authors considered whether calcium dependent mechanisms (i.e. activity dependent TFs) are mediating the expressional/current density changes observed? Would clamping intracellular Ca2+ (i.e. with BAPTA-AM) in the presence of PKC activation augment the expressional changes observed?

       Thank you for your important comments on the introduction section regarding the publications on the T-type Ca2+ channel from our laboratory (Wang et al 2007, Zheng et al 2010). Precisely speaking, these two publications did not handle the maturation of the T-type Ca2+ channel in cardiomyocytes.  The former demonstrated actions of the transcription factor Csx/Nkx2.5 on the Cav3.2 channel, and the latter clarified acute PKC actions on the T-type Ca2+ channel current gating in regard to intracellular Ca2+ concentration. Honestly speaking, at the time of these articles publication, we did not pay much attention to the maturation of the T-type Ca2+ channel in the rat heart during the neonatal period. We did know that the T-type Ca2+ channels were abundant in neonatal ventricular cardiomyocytes and they were extremely sparse in adult ventricular cardiomyocytes. However we did not know that the T-type Ca2+ channels density was getting declined so quickly during the first several days after the birth. Thus in these two publications, neonatal cardiomyocytes were obtained from 1-3 day-old neonatal rat heart; harvesting day of cardiomyocyte was not precisely controlled after the birth.  As you pointed out, and we also believe, that these backgrounds are very important to understand the rationale of the study. Therefore, in this revision, we modified/added some words in relation to the T-type Ca2+ channel maturation in the introduction section. We hope these changes are helpful for the readers of the article to recognize the importance of the signals that govern the channel maturation.  Based on the experiments shown in Figure 8, we speculate that PKCβII may be responsible for the transcription of the transcription factor Nkx2.5. Because conventional PKC isoforms (PKCα, PKCβI, PKCβII and PKCγ) are all dependent on the actions of intracellular [Ca2+], an experiment designed to control the intracellular Ca2+ concentration to assess the expression of Nkx2.5 may be helpful to reconfirm the importance of PKCβII on the regulation of Nkx2.5 expression. However, we did not perform that experiment in this study because of the shortage of time. We would like to ask for your understanding.  We did not perform electrophysiological experiments on ICa.T after long-term modulation of intracellular [Ca2+] in this study, because PMA could not halt the declining of ICa.T density (Figure 4C).  Again we would like to ask for your generous understanding. 

  1. Overexpression of Nkx2.5 was shown to effect the maximum Icat current after 24h. Where there changes observed in cellular automaticity?

       Thank you for picking up that point. We also believe that acceleration of cellular beating (automaticity) by overexpression of Nkx2.5 is the strong evidence that T-type Ca2+ channel plays crucial role for the generation of pacemaker rhythm with response to activation of the transcription by Nkx2.5.  Because those data have already been demonstrated in our previous publication (Wang et al 2007), we refrain from corroborating similar data again in this article.

  1. Is it possible that this study could shed light on Cav3.2 associated expressional changes observed in pathological cardiac hypertrophy?  

         Thank you for referring to the importance of the T-type Ca2+ channel in the pathological condition of the heart.

In response to your suggestion, we have modified the discussion 4.2. regarding the pathological signals including PKC and the contribution of the T-type Ca2+ channel in the heart.

Round 2

Reviewer 1 Report

.

Reviewer 2 Report

The authors have addressed all concerns. The MS is now suitable for publication.